# PeerJ

# A computational method for the systematic screening of reaction barriers in enzymes: searching for *Bacillus circulans* xylanase mutants with greater activity towards a synthetic substrate

Martin R. Hediger[1], Casper Steinmann[2], Luca De Vico[1] and Jan H. Jensen[1]

[1] Department of Chemistry, University of Copenhagen, Copenhagen, Denmark
[2] Department of Physics, Chemistry and Pharmacy, University of Southern Denmark, Odense, Denmark

## ABSTRACT

We present a semi-empirical (PM6-based) computational method for *systematically* estimating the effect of all possible single mutants, within a certain radius of the active site, on the barrier height of an enzymatic reaction. The intent of this method is not a quantitative prediction of the barrier heights, but rather to identify promising mutants for further computational or experimental study. The method is applied to identify promising single and double mutants of *Bacillus circulans* xylanase (BCX) with increased hydrolytic activity for the artificial substrate *ortho*-nitrophenyl $\beta$-xylobioside (ONPX$_2$). The estimated reaction barrier for wild-type (WT) BCX is 18.5 kcal/mol, which is in good agreement with the experimental activation free energy value of 17.0 kcal/mol extracted from the observed $k_{cat}$ using transition state theory (*Joshi et al., 2001*). The PM6 reaction profiles for eight single point mutations are recomputed using FMO-MP2/PCM/6-31G(d) single points. PM6 predicts an increase in barrier height for all eight mutants while FMO predicts an increase for six of the eight mutants. Both methods predict that the largest change in barrier occurs for N35F, where PM6 and FMO predict a 9.0 and 15.8 kcal/mol increase, respectively. We thus conclude that PM6 is sufficiently accurate to identify promising mutants for further study. We prepared a set of all theoretically possible (342) single mutants in which every amino acid of the active site (except for the catalytically active residues E78 and E172) was mutated to every other amino acid. Based on results from the single mutants we construct a set of 111 double mutants consisting of all possible pairs of single mutants with the lowest barrier for a particular position and compute their reaction profile. None of the mutants have, to our knowledge, been prepared experimentally and therefore present experimentally testable predictions.

Corresponding author
Jan H. Jensen, jhjensen@chem.ku.dk

## INTRODUCTION

Rational design of enzyme activity tends to be heuristic in that to varying degrees it is based on inspiration derived from manual inspection of related protein structures (*Patkar et al., 1997*; *Nakagawa et al., 2007*; *Syrén & Hult, 2011*; *Syrén et al., 2012*). One notable exception is the work by Baker and co-workers (*Röthlisberger et al., 2008*; *Jiang et al., 2008*; *Siegel et al., 2010*) in which the desired transition state (TS) is found computationally for a small idealized protein model using quantum mechanical (QM) methods followed by automated optimization of protein scaffold to optimize the affinity to the TS structure and catalytic side chain conformations. While state-of-the-art, this work has not yet led to the design of enzymes that are significantly better than those obtained by conventional means and additional computational approaches may be needed.

We have recently published a computational methodology for directly estimating the effect of mutations on barrier heights (*Hediger et al., 2012b*) and shown that the method is sufficiently fast to screen hundreds of mutants in a reasonable amount of time while also being sufficiently accurate to identify promising mutants (*Hediger et al., 2012a*). As with the methodology developed by Baker and co-workers, the intent of this method is not a quantitative prediction of the barrier heights, but rather to identify promising mutants for further computational or experimental study. Since the method is designed to quickly screen hundreds of mutants several approximations are made: the PM6 semiempirical QM method is used, a relatively small model of the protein is used, and the effect of solvent and structural dynamics is neglected. Furthermore, like most computational studies of enzymatic catalysis, the focus is on estimating $k_{cat}$ rather than $k_{cat}/K_M$. Nevertheless, in an initial application the method was found sufficiently accurate to identify mutations of *Candida antarctica* lipase B with increased amidase activity (*Hediger et al., 2012a*).

This paper presents several improvements to the method: (1) A *systematic* screening of single mutants by automatic generation of all possible single mutations at sites within a certain radius of the active site. (2) Use of the entire protein structure, rather than parts of it. (3) Inclusion of bulk solvent effects through a continuum model.

The method is applied to identify promising single and double mutants of *Bacillus circulans* xylanase (BCX) with increased hydrolytic activity for the artificial substrate *ortho*-nitrophenyl $\beta$-xylobioside (ONPX$_2$). This system was chosen for two reasons: (1) To test the applicability of PM6 to model this general type of chemical reaction. (2) Hydrolysis of ONPX$_2$ by BCX is well-studied (*Joshi et al., 2000*; *Joshi et al., 2001*). Since the focus of this paper is solely the development of computational methodologies, the predicted mutants are therefore presumably amenable to experimental testing by experimental groups.

## METHODS

### Computational details

Most geometry optimizations are performed using PM6 and the molecular orbital localization scheme MOZYME as implemented in MOPAC2012 (*Stewart, 1990*; *Stewart, 1996*; *Stewart, 2007*). From earlier work (*Hediger et al., 2012b*), it was found that the

**Peer**J

orthogonality between the localized molecular orbitals is lost during the geometry optimization and it was suggested to report results only from re-orthogonalized MOZYME single point energy calculations (SPEs). In the current work, however, we find that when doing the SPE calculations, the MOZYME routine frequently fails to generate the same Lewis structure (required for the construction of the localized molecular orbitals) as it did in the start of the geometry optimizations and so the energy from re-orthogonalization is not comparable with the energy of the optimized structure. The implications of this aspect are further discussed below. The reported difficulty arises mainly for structures resembling the transition state, not for stationary points. Thus the convergence of stationary point relative energies, depending on NDDO cutoff and gradient convergence criterion (GNORM), is evaluated based on re-orthogonalization of the MOZYME wavefunction and a NDDO cutoff of 15 Å. Furthermore, the effect of a solvent with dielectric constant $\epsilon_r = 78$ is described by the COSMO model (*Klamt & Schüürmann, 1993*).

Energetic refinement of the MOZYME structures is carried out using the two-body Fragment Molecular Orbital method (FMO2) (*Nakano et al., 2002*; *Fedorov & Kitaura, 2007*) with second order Møller-Plesset perturbation theory for correlation effects (*Fedorov & Kitaura, 2004*), and using the polarizable continuum model (PCM) (*Tomasi, Mennucci & Cammi, 2005*; *Fedorov et al., 2006*) for solvation. All FMO2 calculations are performed using GAMESS (*Schmidt et al., 1993*). Inputs for the FMO2 calculations are prepared using FRAGIT (*Steinmann et al., 2012*). In all FMO2 calculations, the reaction fragment consists of ONP, the first xylose unit and Glu78 in order to keep the reacting species and leaving group in one fragment. This fragment has 45 atoms. In all FMO2 SPEs we use the 6-31G(d) basis set (*Hariharan & Pople, 1973*; *Francl et al., 1982*). Pairs of fragments which are separated by more than two van der Waals radii are calculated using a Coulomb expression for the interaction energy and correlation effects ignored (`resdim=2.0 rcorsd=2.0` in `$fmo`). Optimizations using FMO are carried out with the Frozen Domain and Dimers (FDD) approach (*Fedorov, Alexeev & Kitaura, 2011*) where only residues within 3 Å within the reaction fragment are allowed to relax.

### Estimating the barrier height for the WT

In this study we only model the first, rate-determining (*Joshi et al., 2000*; *Joshi et al., 2001*), step of the mechanism, which is the formation of a glycosyl-enzyme complex (**GE**) from the enzyme substrate complex (**ES**) as illustrated in Fig. 1. The substrate is xylobioside-*ortho*-nitrophenol (ONPX$_2$). The energy barrier is obtained from geometrical interpolation between the two stationary points of the rate-limiting glycosylation step (enzyme-substrate complex, **ES**, and glycosyl-enzyme, **GE**). The various possible sequences of computational steps to obtain the structure of these end points (for both wild-type and mutants) lead to different calculation pathways. In the following, we provide descriptions of the calculation pathways, the implications of which are discussed in the Results section.

**Figure 1 Conventional glycosylation step.** $x_1$: Constrained reaction coordinate; $R$: xylose; $R'$: ortho-nitrophenol (ONP). For discussion of proton transfer from E172 to substrate, see text. $C_1$ indicating nucleophilic carbon of first xylose unit.

In all calculations, the reaction coordinate is defined by the distance between $O^\epsilon$ and the carbon of the first xylose unit bonded to ONP, $x_1$ in Fig. 1. The nucleophilic attack of E78 occurs on the bond between ONP and the first xylose unit. Constraining only a single distance parameter in all potential energy scan calculations helps ensure that synchronicity issues of concerted bond breaking and formation processes are automatically accounted for by the quantum chemical method. As described in our previous studies (*Hediger et al., 2012a*; *Hediger et al., 2012b*), the reaction barrier potential energy is estimated from a linear interpolation procedure. Here the reaction coordinate is frozen to ten intermediate values while the remaining active region is energy minimized to create a reaction profile. In the analysis, the barrier is defined as the highest energy minus the lowest energy, which must be before the highest energy point on the reaction profile. If the last frame of the interpolation has the highest energy, the barrier is not evaluated.

Upon insertion of the substrate in the active site by molecular modeling, the ONP unit is relaxed using molecular mechanics in a fixed enzyme environment. In an extension of the initially proposed approach *Hediger et al., 2012a*; *Hediger et al., 2012b*, not only part of the enzyme but the full enzyme structure is used in the calculations. From this the ambiguity of selecting an appropriate set of residues to model the active site is eliminated.

From careful analysis, we find that the interpolation of the wild-type can be prepared by two slightly different procedures which are illustrated in Fig. 2. The modeling steps start (node "Start" in Fig. 2) with the preparation of the glycosyl-enzyme complex since this structure is conformationally less mobile due to the covalent link.

In the first procedure, called "Interpolation 1", the structure used as input for the optimization of the **GE** ("WT GE" in Fig. 2), with ONP in the active site but not covalently bound to the first xylose unit, is prepared from the crystal structure with PDB ID 1BVV (*Sidhu et al., 1999*). The **ES** complex, "WT ES", is formed by removing the covalent linkage between the substrate and E78 (step "Modify substrate (1)"). The geometry of both structures is optimized (steps "Optimize") without applying any constraints and the resulting structures (referred to as "WT ES opt" and "WT GE opt") are used for interpolation 1.

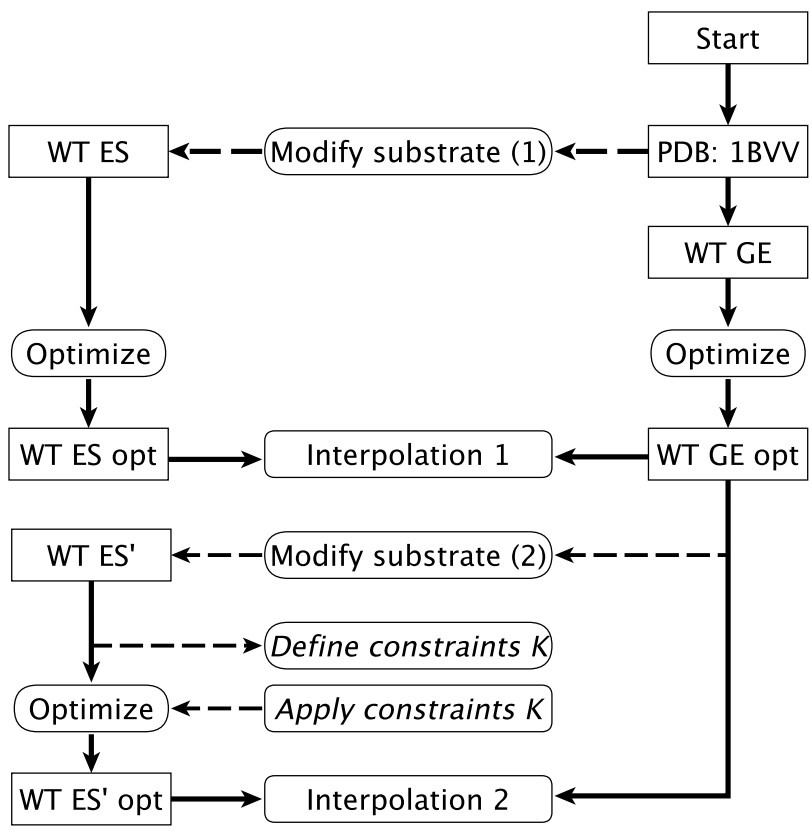

**Figure 2** Calculation pathway for WT interpolations.

In the second procedure, "Interpolation 2", WT GE optimized structure is used as a template for the **ES** complex, referred to as "WT ES'", (the single prime indicating that the structure is derived from a wild-type **GE** structure). The WT ES' structure is again prepared by removing the covalent linkage between the substrate and E78 (step "Modify substrate (2)"). To reduce the computational time required for the geometry optimization, a set of Cartesian constraints $K$ can be defined (step "Define constraints $K$") and applied to spatially fix an outer layer of residues away from the active site (step "Apply constraints $K$"). These constraints are only applied to parts of the enzyme which are already optimized in a preceding step. After optimization of WT ES'', the reaction barrier is mapped out by interpolation 2. Because the structure of WT ES' is optimized to a large degree, the time requirement is greatly reduced and the results are found to be more reliable, see below.

## Estimating barrier heights for the mutants

Three different interpolation procedures for mapping out the reaction barriers of mutants are defined, Fig. 3. In Interpolation 3, the structures WT ES opt and WT GE opt are used in the preparation of the corresponding mutant structures ("Mut ES opt" and "Mut GE opt"), which are used to prepare the interpolation.

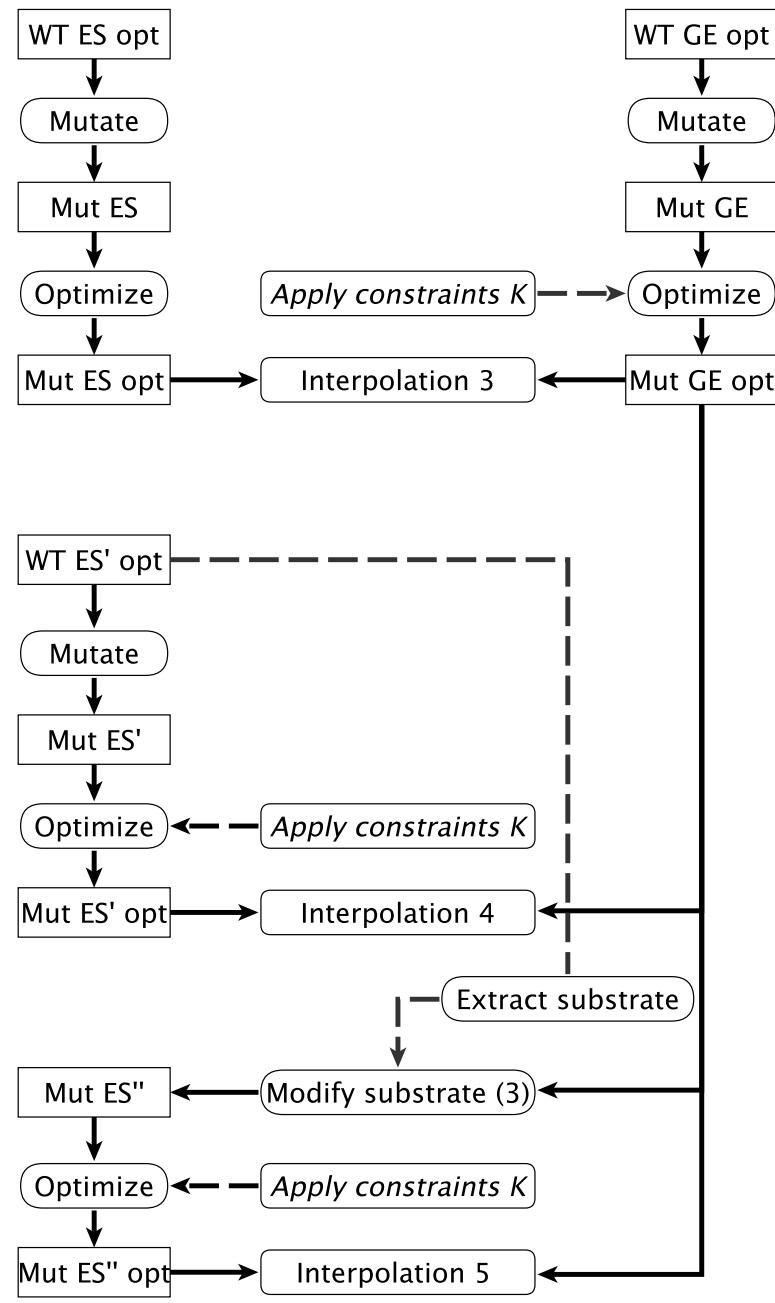

**Figure 3 Calculation pathways for mutant interpolations.** The diagram continues by the nodes "WT ES opt" and "WT GE opt" from Fig. 2.

In Interpolation 4, the structure of the **ES** complex of the mutants is based on the WT ES' structure and is referred to as "Mut ES' opt", the single prime again indicating that the structure is derived from a WT **GE** structure).

In Interpolation 5, the mutant **ES** structures are prepared from the Mut GE opt structure by replacing the covalently bound substrate with the non-covalently bound substrate of

**Table 1 Relative energies [kcal/mol] for different combinations of GNORM and CUTOFF.** Energies obtained as MOZYME^Reortho//MOZYME.

| CUTOFF [Å] | GNORM [kcal/(molÅ)] | | | | | |
|---|---|---|---|---|---|---|
| | 5.0 | 4.0 | 3.0 | 2.0 | 1.0 | 0.5 |
| **ES** | | | | | | |
| 9 | 0.0 | −13.7 | −14.1 | −22.9 | −19.4 | −18.2 |
| 12 | −5.5 | −13.9 | −14.0 | −16.0 | −26.1 | −20.4 |
| 15 | −34.0 | −35.3 | −35.2 | −45.9 | −48.9 | −49.5 |
| **GE** | | | | | | |
| 9 | −3.8 | −7.3 | −10.7 | −14.2 | −21.4 | −19.7 |
| 12 | −10.0 | −10.9 | −24.4 | −24.4 | −26.2 | −24.5 |
| 15 | −9.5 | −25.4 | −27.5 | −27.6 | −43.6 | −40.6 |

WT ES' (steps "Extract substrate" and "Modify substrate (3)"). The mutant **ES** structures are referred to as Mut ES", the double primes indicate that the structure is derived from a mutant **GE** structure (as opposed to being derived from a wild-type **GE** structure). We believe this way of preparing the structure of the **ES** complex of the mutants is most efficient and readily implemented. Other options would be to prepare the **ES** complex by docking procedures which would require considerable effort if hundreds of mutants are to be evaluated. As presented, the operation is a simple matter of command-line scripting.

The molecular structures of the mutant side chains are prepared using the PyMOL (*The PyMOL Molecular Graphics System, 2010*) mutagenesis wizard in combination with local optimization of the mutated side chain using the PyMOL sculpting function.

In Interpolations 4 and 5, to reduce the time demand of the geometry optimizations, optionally the set of constraints $K$ can be applied. Constraints cannot be meaningfully applied in Interpolation 3 because the interpolation between Mut ES and Mut GE produces (prior to being optimized) slightly different input coordinates which, when fixed, result in enormous increases in energy.

## RESULTS AND DISCUSSION

### Stationary points in the WT mechanism

**MOPAC configuration**. The speed and accuracy of MOZYME geometry optimizations are characterized by the gradient convergence criterium (GNORM) and the cutoff distance beyond which NDDO approximations are replaced by point charges (CUTOFF). Table 1 shows the energies of the optimized wild-type **ES** and **GE** for different configurations of MOPAC. The energies are relative to **ES** computed with GNORM = 5.0 kcal/mol and CUTOFF = 9 Å. It is observed that the calculations converge for both **ES** and **GE** when GNORM = 1.0 kcal/(molÅ) and the NDDO cutoff is 15 Å.

Table 2 shows the time requirements for the geometry optimization of **ES** and **GE**. As expected the geometry optimization requires significantly more time when using strict gradient convergence criteria. However this appears to be required in order to obtain converged relative energies. In all of the following, unless otherwise stated, the NDDO cutoff is set to 15 Å and the gradient convergence is 1.0 kcal/(molÅ).

**Table 2** Time requirements [h] for optimizations with different combinations of GNORM and CUT-OFF.

| CUTOFF [Å] | GNORM [kcal/(molÅ)] | | | | | |
|---|---|---|---|---|---|---|
| | 5.0 | 4.0 | 3.0 | 2.0 | 1.0 | 0.5 |
| **ES** | | | | | | |
| 9 | 45.7 | 69.0 | 72.1 | 112.1 | 241.3 | 244.5 |
| 12 | 67.8 | 94.3 | 95.0 | 127.5 | 267.0 | 261.9 |
| 15 | 102.8 | 124.5 | 129.0 | 193.2 | 266.7 | 380.5 |
| **GE** | | | | | | |
| 9 | 53.3 | 57.3 | 76.5 | 111.7 | 147.7 | 156.2 |
| 12 | 69.4 | 64.2 | 104.3 | 117.3 | 176.5 | 174.6 |
| 15 | 92.7 | 110.2 | 144.1 | 198.2 | 411.4 | 424.3 |

## Wild type mechanism and reaction barrier

As described in the Methods section, the enzyme substrate complex for the WT is constructed in two ways leading to two different interpolation procedures: "Interpolation 1" and "Interpolation 2" shown in Fig. 2. Interpolation 1 yields an irregularly shaped reaction profile (Fig. S1) from which it is impossible to extract a reaction barrier. Unconstrained Interpolation 2 yields a reasonable looking reaction profile (Fig. 4A) with a barrier of 18.5 kcal/mol, which is in agreement with the experimental activation free energy value of 17.0 kcal/mol extracted from the observed $k_{cat}$ *Joshi et al. (2000)* using transition state theory. If the constraints $K$ are not applied, the geometry optimization at each interpolation point along the reaction profile requires between 100 and 300 CPU hours (Fig. 4B), a prohibitive cost if hundreds of mutants are to be screened. The CPU time requirement can be reduced to less than 50 CPU hours by only optimizing the geometry of residues close to the active site (Fig. 4B) and freezing the rest of the coordinates to their values in the optimized **GE** complex. Optimizing only those residues within 8, 10 and 12 Å of the active site (OPT 8 Å, OPT 10 Å, OPT 12 Å in Fig. 4) reduces the predicted barrier to 10.0, 13.4 and 14.4 kcal/mol respectively (Fig. 4A). Much of this effect will likely cancel when barriers for mutants are compared to WT but, based on these results, it is advisable to recompute the barriers of the most promising mutants without constraints. This is further discussed below.

Interestingly, for the **GE** intermediate the proton is found to reside on E172 rather than ONP as in the canonical mechanism (Fig. 1) with hydrogen bonds to both the phenol oxygen and one of the oxygen atoms on the nitro group (Fig. 5). A corresponding stationary point with a protonated ONP group does not appear to exist. Geometry optimizations using FMO-MP2/6-31G(d):RHF show that there is a stationary point both with protonated ONP and protonated E172 contrary to the findings by PM6 which is in line with the canonical mechanism. It is therefore likely that the deprotonated ONP dissociates first followed by deprotonation of E172. Removal of the nitro-group leads to proton transfer to the phenol group so this issue likely only applies to the ONPX$_2$ substrate.

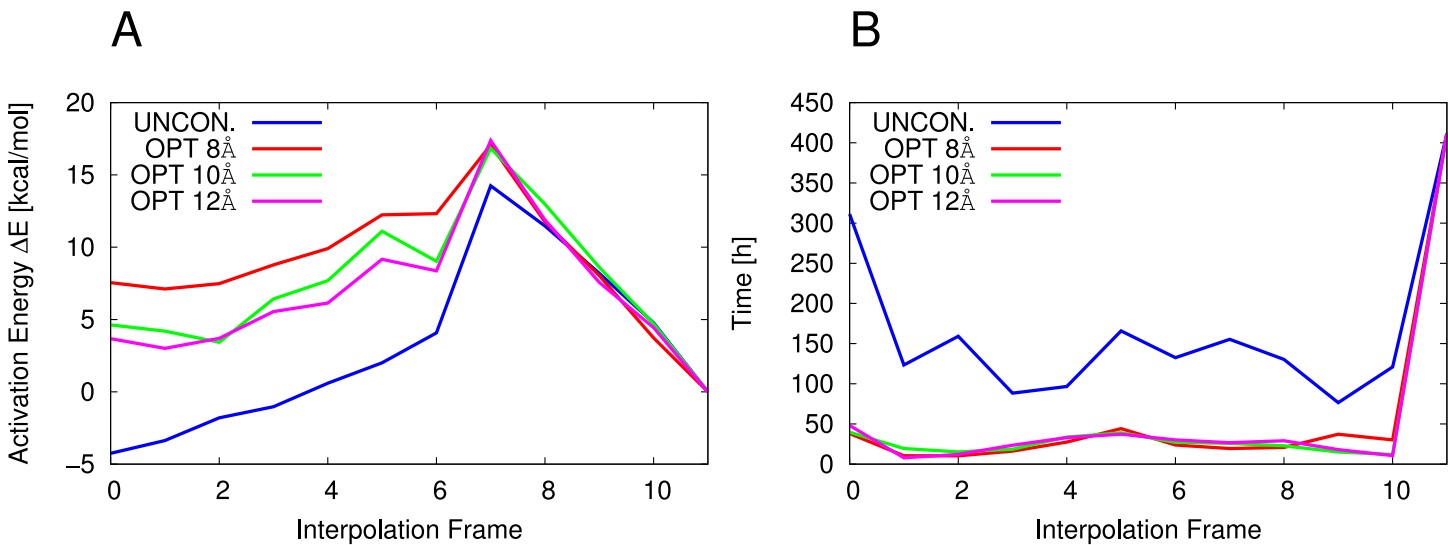

**Figure 4 WT reaction barrier and time requirements.** Interpolation 2, $\varepsilon_r = 78$, UNCON. = Unconstrained and constrained optimizations, No constraints applied in optimization, $\Delta E$ calculated using MOZYME, GNORM = 1.0 kcal/(mol Å), NDDO CUTOFF = 15 Å. (A) Reaction barriers. (B) Time requirements.

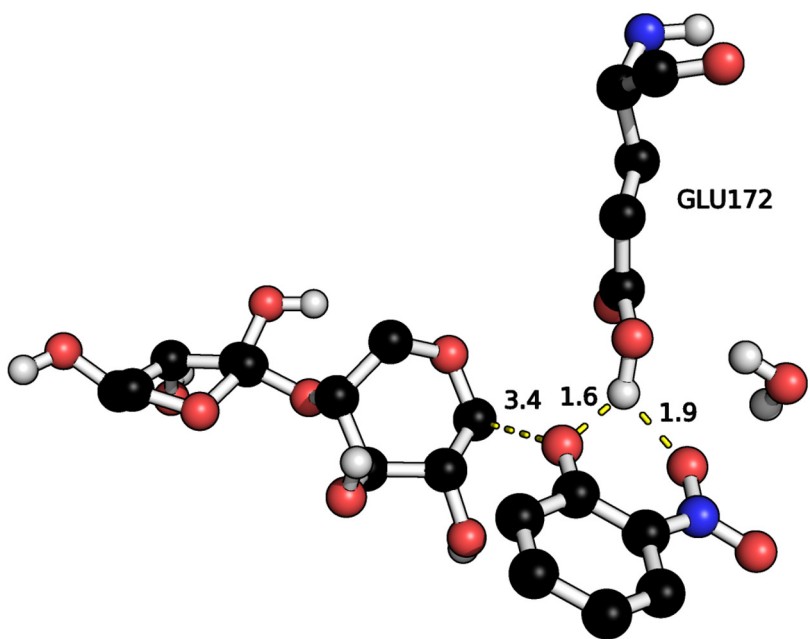

**Figure 5 Hydrogen bonds between ONP leaving group and E172 proton in the optimized GE.** Distances in Å.

## Interpolation schemes for mutants

Three interpolation schemes are tested for predicting reaction profiles of mutants as outlined in the Methods section and Fig. 3.

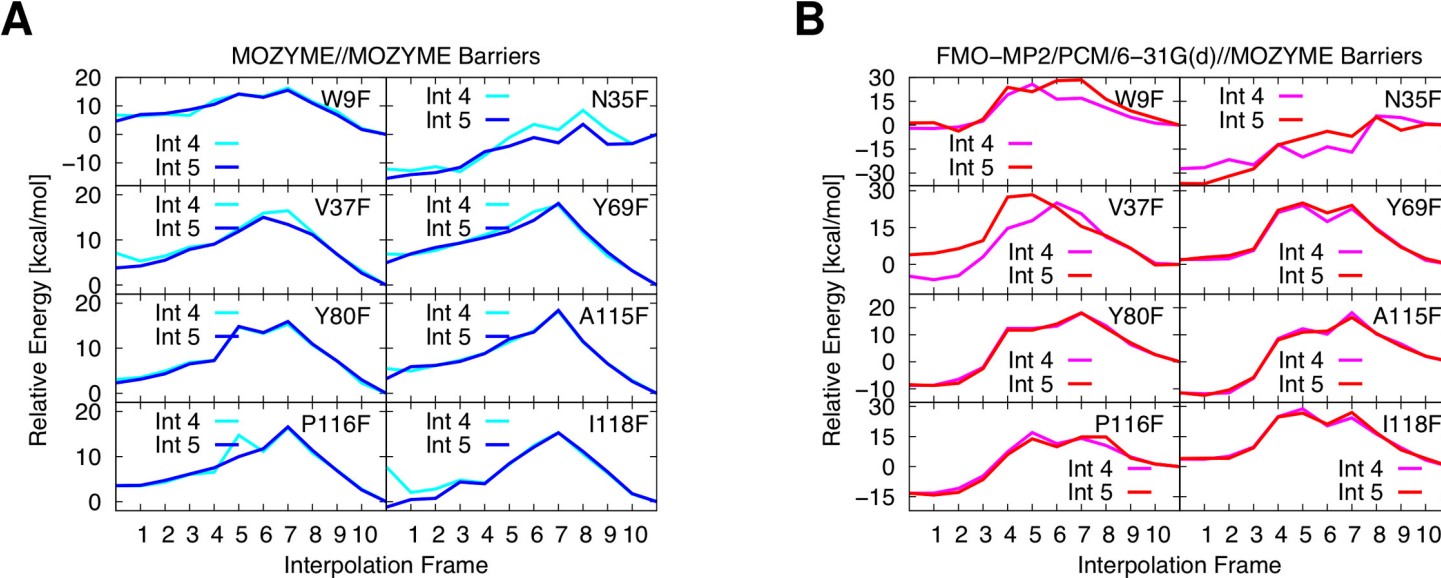

**Figure 6 Constrained interpolations 4/5, optimized layer: 8 Å.** (A) Barriers from MOZYME optimized structures. (B) FMO/PCM barriers based on SPE calculations of the MOZYME optimized structures.

**Interpolation 3.** Interpolation 3 is most closely related to Interpolation 1 for the WT and is tested for six single point mutations where coordinates of all residues within 8 Å of the active site are optimized. Like for the WT, this approach leads to irregularly shaped reaction profiles from which it is impossible to extract reaction barriers (Fig. S2).

**Interpolations 4 and 5.** Interpolations 4 and 5 differ on whether the mutant **ES** structure is constructed from the WT **ES** structure (Interpolation 4) or the mutated **GE** structure (Interpolation 5). Both approaches are tested for eight single point mutations where the coordinates of all residues within 8 Å of the substrate are optimized. The mutations are all within the active site and close proximity to the ONP leaving group and E172. For the studied mutants, we find that all reaction profiles appear conclusive in shape and readily permit the estimation of a barrier height (Fig. 6).

As shown in Fig. 7, the required time to calculate the barriers is mostly within the desired time frame of two days when using Interpolation procedure 5.

The time requirements for Interpolation 4 are found to be higher.

Furthermore, since two "Mutate"-modeling steps are involved in Interpolation 4 (Fig. 3), local optimization of the mutant side chain during the modeling process can result in differently oriented side chains for the **ES** complex and **GE** intermediate of the mutant leading to non-physical structures in the interpolation procedure. Interpolation 5 is in this sense more robust in that all mutated side chains, by definition of the interpolation procedure, are identically oriented in both the **ES** and **GE** structures.

The PM6 reaction profiles shown in Fig. 6A are recomputed using FMO-MP2/PCM/6-31G(d) single points as shown in Fig. 6B. We confine our comparison to the Interpolation 5 results as this is the scheme we will use for the remaining mutants. The changes in barrier heights relative to WT computed with PM6 compare well with the corresponding FMO

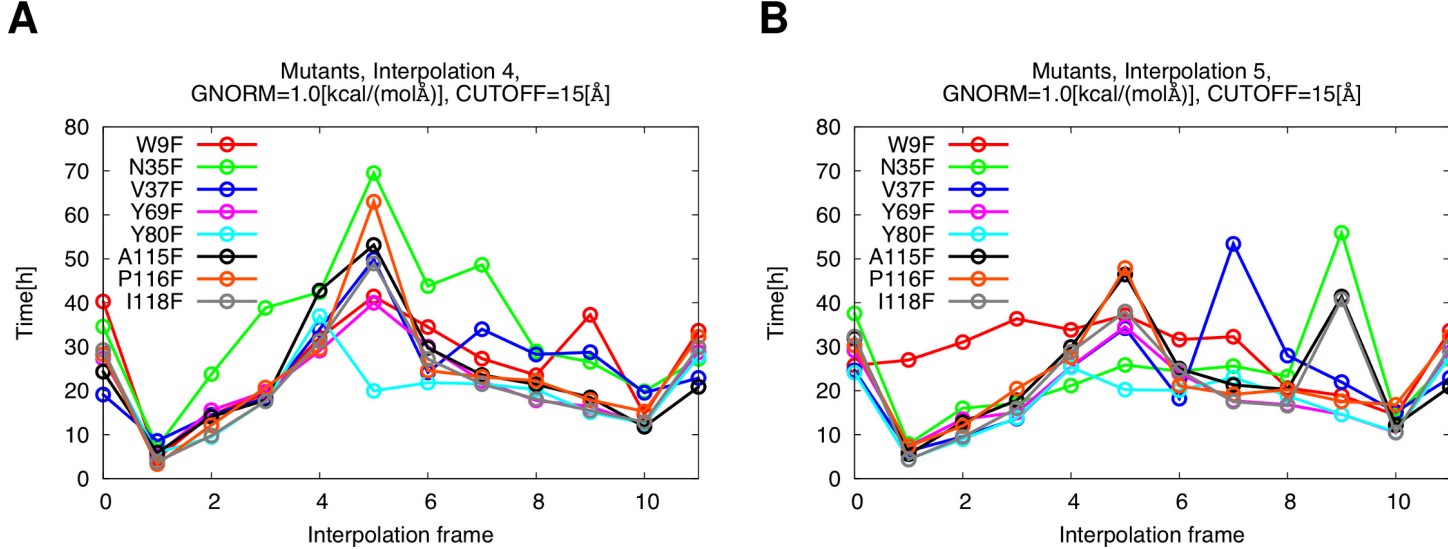

**Figure 7** **Time requirements, optimized layer of residues: 8 Å.** (A) Interpolation procedure 4. (B) Interpolation procedure 5.

values with an average error of $0.7 \pm 5.3$ kcal/mol. The largest error (9.5 kcal/mol) occurs for I118F for which PM6 predicts a barrier height increase of 6.5 kcal/mol while FMO predicts a 3.0 kcal/mol decrease. More qualitatively, PM6 predicts an increase in barrier height for all eight mutants while FMO predicts an increase for six of the eight mutants. Both methods predict that the largest change in barrier occurs for N35F, where PM6 and FMO predict a 9.0 and 15.8 kcal/mol increase, respectively. We thus conclude that PM6 is sufficiently accurate to identify promising mutants for further study.

## Computational high throughput screening of BCX mutants

We prepared a set of all theoretically possible (342) single mutants in which every amino acid of the active site (except for the catalytically active ones E78 and E172) was mutated to every other amino acid. The active site is defined as every residue that has at least one atom within 4 Å of the substrate.

However it was not possible to calculate the reaction barrier for every mutant because in some cases the modeling procedure of the stationary points resulted in geometries for which MOPAC cannot generate a Lewis structure or because MOPAC predicts a wrong total charge. In case MOPAC is unable to generate a Lewis structure, it is not possible to start the calculation and so these mutants are identified when the calculations are submitted. To check for correct computation of total charge, we use a computer script which compares the value found by MOPAC, using the CHARGES keyword, with the true value assuming standard protonation of all ionizable residues. We have made no attempt to fix these calculations but simply discard them from the analysis. Subsequent visual inspection of the mutants for which the calculation did not start reveals that this was only the case when the newly introduced side chain is a proline or a tyrosine and when the environment is very compact. To model the side chain conformations, we use the PyMOL modeling and mutagenesis routines and also apply a local optimization of the mutated side

chain, keeping the environment fixed. The PyMOL modeling routine only optimizes bond lengths, angles and interatomic distances but does not consider electrostatic or electronic effects. In the case of proline we observe that the ring can be greatly distorted and in case of tyrosine the ring can be distorted to a boat conformation when trying to place it in a sterically congested environment. An additional reason for not being able to calculate the reaction barrier is that in some cases, one or more side chains in the **ES** complex are oriented significantly different from the **GE**. In such cases, when the interpolation frames are prepared, it can happen that two atoms are placed at very short distances to each other and MOPAC will not start the calculation again for such a structure. Furthermore, we discarded a number of double mutants if the optimization required more than five days. All discarded mutants are listed in Tables S1 and S2. We note that the modeling of side chains by the presented approach is a fast and simple procedure which can be improved by using more elaborate techniques of modeling the side chains (e.g., by molecular mechanics optimizations) and which is the subject of future studies.

Finally, the reaction barrier was calculated for 317 single and 111 double mutants using an optimization layer of 10 Å.

The calculated barriers are found to be mostly independent of reorthogonalization of the wavefunction of the converged geometry, and the qualitative conclusions (lower/higher barrier compared to wild-type) are the same in 80% of the cases with barriers lower than 34 kcal/mol, Fig. S3. Based on this observation and on the above discussion, we therefore consider energies obtained without reorthogonalization.

The 20 single mutants with the lowest barriers are listed in Table 3. All barriers are lower than the WT value, which is 13.4 kcal/mol for the 10 Å optimization. Based on results from the single mutants we construct a set of double mutants consisting of all possible pairs of single mutants with the lowest barrier for a particular position, using the same set of constraints $K$ as for the single mutants. Just as for the single mutants the PyMOL construction of some side chains resulted in unphysical structures which were discarded from the analysis using the criteria discussed for the single mutants. Furthermore, in some cases the optimization of some points on the reaction profiles of a double mutant failed to converge after five days of CPU time and so the corresponding mutant was discarded as well. The average time for optimization over all interpolation frames of double mutants is observed to be 29 h. In total, the barriers for 111 double mutants are calculated and the lowest twenty barriers for all single and double mutants are listed in Table 3.

An analysis of the distribution of single and double mutant barriers indicates that the effects of single mutations on the barriers are additive and contribute to a lowering of barriers on average, which is shown in Fig. 8.

As discussed above, using the constraints on part of the enzyme decreases the computed barrier for the WT by 5.1 kcal/mol (from 18.5 to 13.4 kcal/mol) if only residues within 10 Å of the active site are optimized (Fig. 4). The assumption is that the *relative* barriers will be less affected, but ideally the barriers of the most promising mutants listed in Table 3 should be recomputed without constraints.

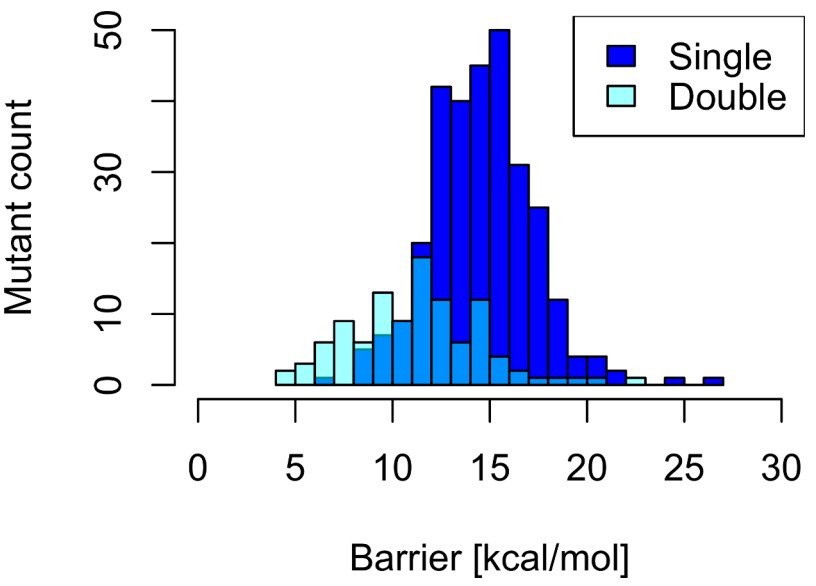

**Figure 8 Barrier distribution of single and double mutants.** Only datapoints below 30 kcal/mol shown.

**Table 3 Twenty single and double mutants with lowest barriers.**

| Reaction barriers [kcal/mol] | | | |
|---|---|---|---|
| **Single mutants** | | **Double mutants** | |
| Q127W | 6.9 | Q7W-Q127W | 4.6 |
| S117P | 8.5 | W9E-Q127W | 5.0 |
| Q127K | 8.6 | Q127W-Y166V | 5.0 |
| A115I | 8.7 | W9E-Y65R | 5.3 |
| Q7W | 8.7 | Q7W-N35E | 5.7 |
| Q7R | 8.8 | N35E-Q127W | 6.2 |
| W9E | 9.1 | V37T-F125K | 6.3 |
| Q127H | 9.5 | W9E-F125K | 6.4 |
| N35E | 9.6 | Q7W-W129I | 6.5 |
| F125K | 9.6 | V37T-Q127W | 6.7 |
| Q127T | 9.6 | W9E-A115I | 6.9 |
| Q127I | 9.6 | Q127W-W129I | 7.1 |
| Q127V | 9.9 | P116C-Q127W | 7.2 |
| F125E | 10.2 | I118M-F125K | 7.2 |
| A115D | 10.3 | Q7W-Y65R | 7.4 |
| Q127L | 10.3 | F125K-Y174D | 7.4 |
| W9F | 10.4 | W9E-Y69E | 7.5 |
| Q127S | 10.6 | A115I-I118M | 7.5 |
| Q127F | 10.6 | I118M-Q127W | 7.6 |
| W9D | 10.7 | F125K-Q127W | 7.8 |

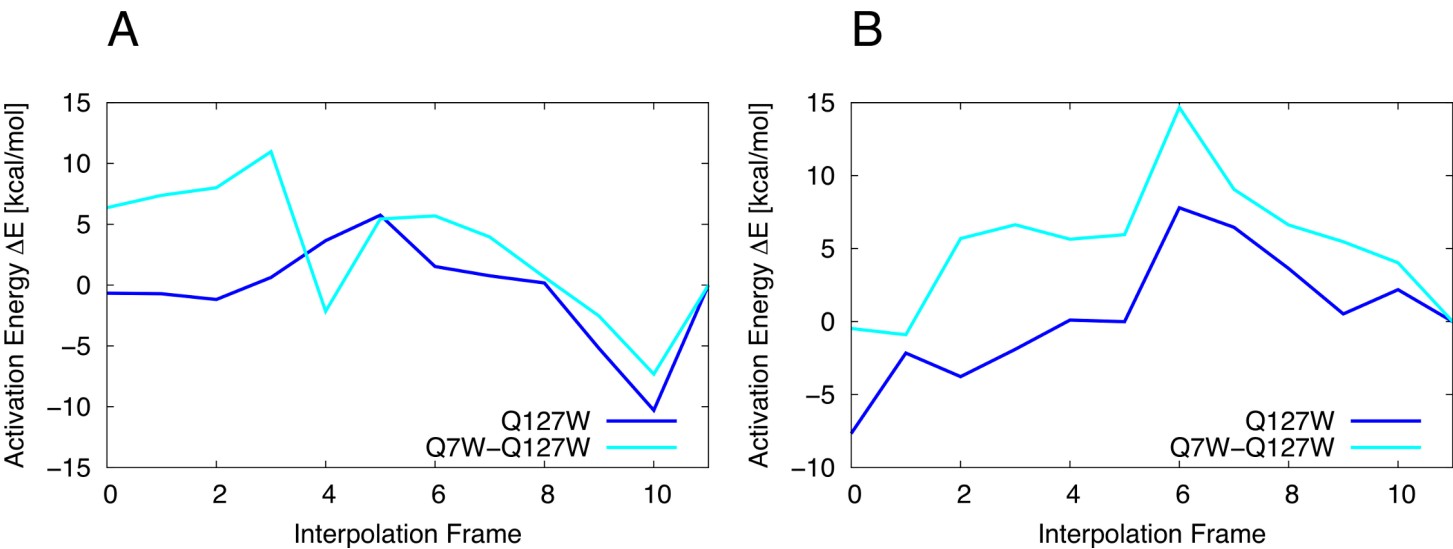

**Figure 9 Barriers of best single and double mutant.** (A) Constrained optimization. Q127W: 6.9, Q7W-Q127W: 4.6 kcal/mol. (B) Unconstrained optimization. Q127W: 15.5, Q7W-Q127W: 15.6 kcal/mol. The starting geometries for these optimizations are the structures optimized with constraints.

Recomputing the reaction barriers of the best single and double mutants (Q127W and Q7W-Q127W) without any applied constraints reveals that, as expected from the convergence study reported in Fig. 4A, the barriers increase (by 8.6 and 11.0 kcal/mol, respectively) but remain below the barrier obtained from the unconstrained WT optimization. The constrained and unconstrained barriers are shown in Fig. 9. This provides further evidence that these mutants indeed react faster than the WT and should be considered for further experimental evaluation. However, the computational cost is significantly increased. The average time to optimize all structures of the constrained interpolation is 28 (Q127W) and 37 (Q7W-Q127W) hours while the average time to optimize all structures without any constraints is 110 (Q127W) and 124 (Q7W-Q127W) hours per single processor (MOPAC2012 is not parallelized). So recomputing the barriers of all 40 mutants listed in Table 3 would require a significant investment of computer time. Alternatively, one could use the constrained geometries as a starting point for conventional QM/MM calculations with *ab initio* QM, at which point dynamical averaging could also be introduced. However, given the time requirements associated with such an approach one might also consider going straight for an experimental verification for unequivocal answers. Either way, the key intent of the method is as an additional tool for generating ideas for possible mutants that other, heuristic, approaches may miss.

While a complete discussion of all mutants listed in Table 3 is beyond the scope of this paper, we provide a rationalization of a few exemplifying mutants in the following. These cases represent different design strategies such as enzyme-substrate complex destabilization or transition state stabilization.

As stated above, the single mutant with the lowest barrier is found to be Q127W. An inspection of the structure shows that the Gln residue in the WT is likely stabilizing the

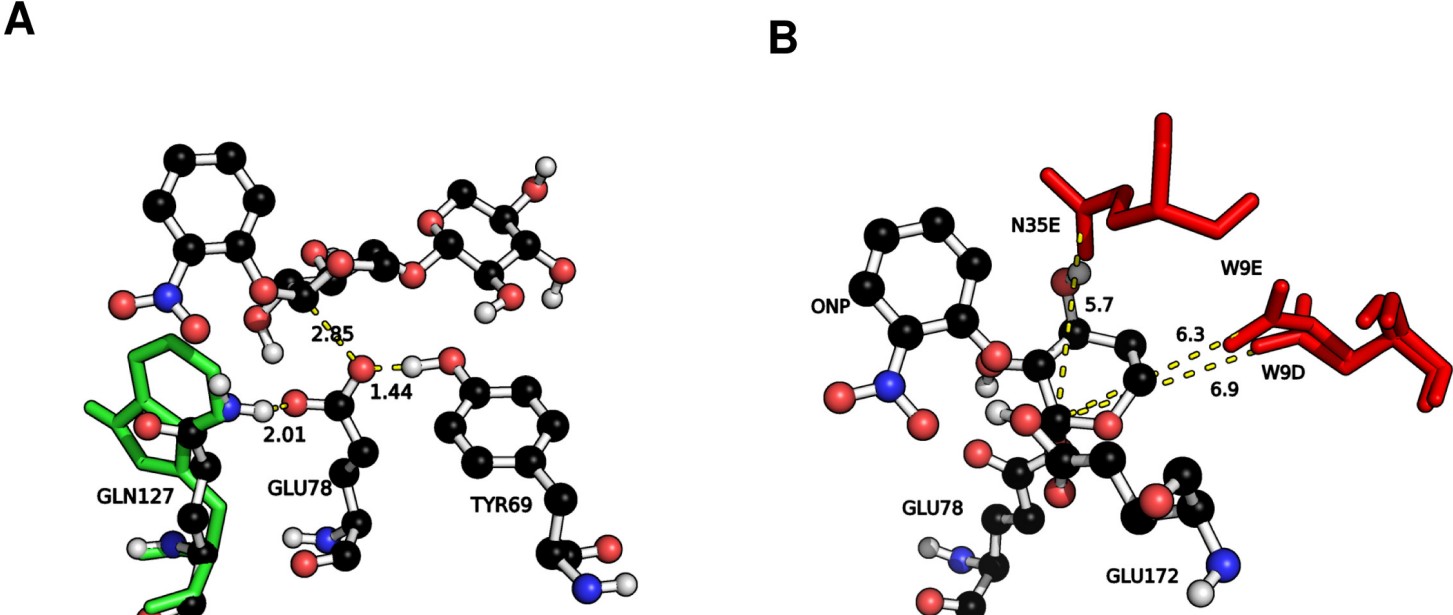

**Figure 10 Rationalization of reaction barriers.** (A) Overlay of WT (black carbon spheres) and Q127W side-chain (green sticks) **ES** complex structures. (B) Coulombic interactions between (negative, red sticks) W9D/E, N35E and the nucleophilic carbon ($C_1$) on the substrate. Distances in Å.

negative charge on E78 in the enzyme-substrate complex while removal of the hydrogen bond donor, Fig. 10A, and replacement by Trp, which is of similar size (allowing to preserve structural integrity of the region), will likely increase the energy of the enzyme-substrate complex and so lower the reaction barrier of the first step. However, with this mutation there is a danger that it will raise the barrier for the second step of the mechanism where negative E78 is regenerated and so careful assessment of the full reaction cycle would be required to fully characterize the effects of the mutation on the total reaction.

In terms of transition state stabilization, it is observed that the mutants W9D and W9E provide favourable Coulombic interactions with the partial positive charge on the nucleophilic carbon of the first xylose unit ($C_1$, Fig. 1) developing during the glycosylation, Fig. 10B. This interaction is likely to stabilize the transition state, compared to WT, and so provides a lowering of the reaction barrier.

To the best of our knowledge, none of the mutations listed in Table 3 have been tested experimentally and can thus be considered predictions. N35D has been shown experimentally to have a larger $k_{cat}$ than WT using the ONPX$_2$ substrate (14.5 *vs.* 9.6 s$^{-1}$ for the WT *Joshi et al., 2000*). The calculated barrier for N35D is 17.6 kcal/mol and considerably higher than the WT. However, *Joshi et al. (2000)* have presented evidence for D35 being protonated at the pH of interest, while our screening method only considers standard protonation states for ionizable residues. Extending the automated screening method to non-standard protonation states is considerably more complicated and a subject for future studies.

## CONCLUSIONS

We present a computational method for *systematically* estimating the effect of all possible single mutants, within a certain radius of the active site, on the barrier height of an enzymatic reaction. The intent of this method is not a quantitative prediction of the barrier heights, but rather to identify promising mutants for further computational or experimental study.

Since the method is designed to quickly screen hundreds of mutants several approximations are made: the PM6 semi-empirical quantum mechanical method is used, the transition state structure is estimated, and the effect of vibrational and structural dynamics is neglected. Furthermore, like most computational studies of enzymatic catalysis, the focus is on estimating $k_{cat}$ rather than $k_{cat}/K_M$. Nevertheless, in an initial application the method was found sufficiently accurate to identify mutations of *Candida antarctica* lipase B with increased amidase activity (*Hediger et al., 2012a*).

The method is applied to identify promising single and double mutants of *Bacillus circulans* xylanase (BCX) with increased hydrolytic activity for the artificial substrate *ortho*-nitrophenyl $\beta$-xylobioside (ONPX$_2$). Since the focus of this paper is solely the development of computational methodologies, the predicted mutants are therefore presumably amenable to experimental testing by experimental groups.

The estimated reaction barrier for wild-type (WT) BCX is 18.5 kcal/mol, which is in agreement with the experimental activation free energy value of 17.0 kcal/mol extracted from the observed $k_{cat}$ (*Joshi et al., 2000*) using transition state theory. The rate determining step is the formation of a glycosyl intermediate **GE** starting with the enzyme-substrate complex **ES**. However, the geometry optimization at each interpolation point along the reaction profile requires between 100 and 300 CPU hours (Fig. 4B), a prohibitive cost if hundreds of mutants are to be screened. The CPU time requirement can be reduced to less than 50 CPU hours by only optimizing the geometry of residues within 10 Å of the active site (Fig. 4B) and freezing the rest of the coordinates to their values in the optimized **GE** complex. While this decreases the reaction barrier (Fig. 4A) by up to 8.5 kcal/mol, we show for a few mutants that this effect partially cancels when applied to changes in barrier height so that promising mutants identified with constraints remain promising after the constraints have been removed.

The PM6 reaction profiles for eight single point mutations are recomputed using FMO-MP2/PCM/6-31G(d) single points as shown in Fig. 6B. PM6 predicts an increase in barrier height for all eight mutants while FMO predicts an increase for six of the eight mutants. Both methods predict that the largest change in barrier occurs for N35F, where PM6 and FMO predict a 9.0 and 15.8 kcal/mol increase, respectively. We thus conclude that PM6 is sufficiently accurate to identify promising mutants for further study. The quality of the calculated reaction barriers is likely improved by including additional effects such as structural dynamics, which are the subject of future studies.

We prepared a set of all theoretically possible (342) single mutants in which every amino acid of the active site (except for the catalytically active residues E78 and E172) was mutated to every other amino acid. The active site is defined as every residue that has

at least one atom within 4 Å of the substrate. Twenty-five of these single mutations were discarded due to steric strain and similar reasons and the reaction profiles were computed for the remaining 317 mutants. Based on results from the single mutants we construct a set of 111 double mutants consisting of all possible pairs of single mutants with the lowest barrier for a particular position and compute their reaction profile. The twenty single and double mutants with lowest barriers are listed in Table 3. The average time for optimization over all interpolation frames of double mutants is observed to be 29 h.

To the best of our knowledge, only two of the mutations listed in Table 3 (Q127H and N35E) have been constructed and $k_{cat}$ has been measured only for N35E (*Ludwiczek et al., 2013*). The measured $k_{cat}$ is lower than WT by a factor of 0.16, and thus not in agreement with our predictions. All but N35E can thus be considered predictions. We hope to be able to verify the suggested mutants in future experimental studies. Alternatively, one could use the constrained geometries as a starting point for conventional QM/MM calculations with *ab initio* QM, at which point dynamical averaging could also be introduced. However, given the time requirements associated with such an approach one might also consider going straight for an experimental verification for unequivocal answers. Either way, the key intent of the method is as an additional tool for generating ideas for possible mutants that other, heuristic, approaches may miss.

### Funding
Computational resources were provided by the Danish Center for Scientific Computing (DCSC). The work was funded in part by the EU through the *in silico* Rational Engineering of Novel Enzymes (IRENE) project. The funders had no role in study design, data collection and analysis, decision to publish, or preparation of the manuscript.

### Grant Disclosures
The following grant information was disclosed by the authors:
EU: the *in silico* Rational Engineering of Novel Enzymes (IRENE) project.

### Competing Interests
We declare no competing interests.

### Author Contributions
- Martin R. Hediger and Casper Steinmann conceived and designed the experiments, performed the experiments, analyzed the data, contributed reagents/materials/analysis tools, wrote the paper.
- Luca De Vico conceived and designed the experiments, analyzed the data, discussion of results.
- Jan H. Jensen conceived and designed the experiments, analyzed the data, wrote the paper.

## Supplemental Information

Supplemental information for this article can be found online at http://dx.doi.org/ 10.7717/peerj.111.

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
