# Peer review of "A computational method for the systematic screening of reaction barriers in enzymes: searching for Bacillus circulans xylanase mutants with greater activity towards a synthetic substrate"

_PeerJ, doi:10.7717/peerj.111_

## Round 0.1 · original submission · Minor Revisions

The comments from Review 1 seems to be concerned with the constraints of the methodology, but he still considers a suitable and interesting method and recommends accept. As such, the paper is acceptable, but I would like to give you the chance to discuss these issues in response to the review if you would like.

·

Basic reporting

The article consists in a revision of the method previously published on ref 10 from (almost) the same authors, in which several modifications are introduced in the generation of the different mutants, and the proposed methodology has been applied to a different biological system.
All the figures and tables are well presented and clarify the content of he article.

Experimental design

My concerns about the method rely on the way the Potential Energy Surface (PES) is explored. Authors use an interpolation algorithm which combines both reactant and product states and then combine the results with consequent optimizations. After, they take the energy corresponding to reactants and compare with the maximum of energy of the interpolated structures. This procedure can introduce large errors in the estimation of the activation barriers depending upon the degree of synchronicity of the chemical step: the degree of bond breaking and bond formation could not be the same, as for example in a step-wise reaction.
Furthermore, evaluation of high level Single Point Energy (SPE) calculations over low level (semi-empirical methods as PM6) obtained geometries can again mislead the position of the transition states over the PES (for an example see J. Phys. Chem. B 2006, 110, 17663).
Thus, i believe that the availability of the transition state (TS) geometry its crucial, in order to perform pondered interpolations between the reactant and product states. This information can be obtained from direct TS search, from methods such as self avoiding walk (SAW) or nudged elastic band (NEB), or direct PES exploration (by means of scanning the distances of the bonds involved in the chemical process).
I personally believe that the PES exploration should be the most reliable and faster one (as long as the precise nature of the TS is not needed, but the evolution of the bonds are), mostly using semi-empiprical methods, and afterwards this PES can be corrected by mean of high level SPE calculations (of a less dense number of points from the low level surface and then extended by means of cubic or akima spline methods).

Additionally, regarding the methodology to produce the mutations (page 12), although i am not aware of the algorithms behind the PyMOL mutagenesis routines, i believe that an small MM optimization of the mutated residue could overcome the problem of obtaining a good guess for MOPAC (mostly with proline and tyrosine mutants as the authors state).

Finally, i would be very glad to see further revisions of the proposed methodology incorporating structure dynamics in the protocol (may be parallelizing the generation of each single mutation) in order to incorporate more flexibility if the final conformation of the active site.

Validity of the findings

Within the limitations of the methodology, the results obtained are well suited and discussed along in the article.
Fast methodologies of this kind are very interesting and can provide useful tips and insides of biochemical processes.

Reviewer 2 ·

Basic reporting

Fine as is.

Experimental design

Experimental validations would be nice.

Validity of the findings

Computations are fine only missing experimental validations.

Additional comments

Using computational means the authors have examined single and double mutants of Bacillus circulans xylanase to identify substitutions that affect the activity of this enzyme towards a non-standard substrate. The computations are fine and appear well validated. The only missing feature is experimental validation.

---

## Round 0.2 · accepted · Accept

Congratulations! I look forward to seeing it in print.